# Structure-Dependent Toxicokinetics of Selected Pyrrolizidine Alkaloids In Vitro

**DOI:** 10.3390/ijms23169214

**Published:** 2022-08-16

**Authors:** Julia Buchmueller, Florian Kaltner, Christoph Gottschalk, Maria Maares, Albert Braeuning, Stefanie Hessel-Pras

**Affiliations:** 1German Federal Institute for Risk Assessment, Max-Dohrn-Str. 8-10, 10589 Berlin, Germany; 2Chair of Food Safety, Veterinary Faculty, Ludwig-Maximilians-Universität München, Schoenleutnerstr. 8, 85764 Oberschleissheim, Germany; 3Institute of Food Chemistry and Food Biotechnology, Justus Liebig University of Giessen, 35392 Giessen, Germany; 4Institute of Food Chemistry and Toxicology, Technische Universität Berlin, Straße des 17. Juni 135, 10623 Berlin, Germany

**Keywords:** pyrrolizidine alkaloids, metabolism, structure-dependency, uptake

## Abstract

Phytochemicals like pyrrolizidine alkaloids (PAs) can affect the health of humans and animals. PAs can occur for example in tea, honey or herbs. Some PAs are known to be cytotoxic, genotoxic, and carcinogenic. Upon intake of high amounts, hepatotoxic and pneumotoxic effects were observed in humans. This study aims to elucidate different toxicokinetic parameters like the uptake of PAs and their metabolism with in vitro models. We examined the transport rates of differently structured PAs (monoester, open-chained diester, cyclic diester) over a model of the intestinal barrier. After passing the intestinal barrier, PAs reach the liver, where they are metabolized into partially instable electrophilic metabolites interacting with nucleophilic centers. We investigated this process by the usage of human liver, intestinal, and lung microsomal preparations for incubation with different PAs. These results are completed with the detection of apoptosis as indicator for bioactivation of the PAs. Our results show a structure-dependent passage of PAs over the intestinal barrier. PAs are structure-dependently metabolized by liver microsomes and, to a smaller extent, by lung microsomes. The detection of apoptosis of A549 cells treated with lasiocarpine and monocrotaline following bioactivation by human liver or lung microsomes underlines this result. Conclusively, our results help to shape the picture of PA toxicokinetics which could further improve the knowledge of molecular processes leading to observed effects of PAs in vivo.

## 1. Introduction

Pyrrolizidine alkaloids (PAs) are a large group of phytochemicals with more than 660 different chemical structures identified, synthesized by a wide variety of plants. PA-producing plants are, for example, Boraginaceae, Asteraceae or Fabaceae [1,2]. Humans are exposed to PAs via the plants themselves or via contaminated food. Prominent exposure sources are tea, leafy vegetables, herbs, plant-based food supplements or honey [3].

All PAs share a common structure of a necine base esterified with one or two necic acids [1]. It is generally accepted that 1,2-unsaturated PAs can exert toxic effects to humans and animals [1,4,5]. Several cases of intoxications after uptake of high doses of PAs were observed. Symptoms like ascites, liver fibrosis, and liver cirrhosis were described as well as hepatomegaly, and the hepatic sinusoidal obstruction syndrome (HSOS) [6,7,8]. PAs exhibit cytotoxic, genotoxic, and carcinogenic properties [1,4,9]. To exert toxicity, PAs require bioactivation comprising enzymatic oxidation to reactive metabolites. These intermediates can spontaneously be dehydrated, resulting in reactive pyrrolic esters that interact with nucleophilic counterparts like proteins or DNA [10,11]. This reaction happens mainly in the liver, which explains why most PA-caused effects are detected here. Nevertheless, the lung can also be affected by PAs. It is well accepted that some PAs induce pulmonary arterial hypertension (PAH) [12,13]. However, PA-mediated toxicity is a consequence of an interplay of many more factors than metabolism. Yang et al. (2001) investigated the oral absorption of PAs in rats and showed that PAs are passively absorbed [14]. However, for the PAs senecionine *N*-oxide, retrorsine *N*-oxide and lycopsamine *N*-oxide, an additional efflux mechanism was detected. A similar effect was investigated by Hessel et al. (2014). They observed that echimidine and heliotrine are transported back into the intestinal lumen by an active P-glycoprotein (ABCB1)-mediated efflux mechanism [15]. After passing the intestinal epithelium, the PAs are transported to the liver. PAs are actively taken up by human hepatic HepaRG cells. Thereby, diester PAs demonstrated higher uptake rates than the examined monoester PAs [16]. Moreover, the knockdown of two liver influx transporters (SLC10A1 and SLC22A1) reduced the uptake of retrorsine into human HepaRG cells compared to control cells without transporter knockdown indicating a considerable influence of liver transporters in PA-mediated toxicity [17]. The role of the uptake transporter solute carrier (Slc)22a1 (also known as Oct1) in transporting retrorsine into rat hepatocytes and SLC22A1-overexpressing Madin–Darby canine kidney cells (MDCK)-cells was also examined by Tu et al. (2014) The results indicate that the transporter plays an important role in transporting retrorsine [18]. Monocrotaline was also demonstrated to be transported by Slc22a1 into MDCK cells [19]. In the hepatocytes, PAs are structure-dependently bioactivated by cytochrome P450 (CYP) monooxygenases [20,21,22].

Taken together, PA-mediated toxicity depends on a variety of factors comprising oral bioavailability, different influx and efflux transport processes in the intestine and liver, structure-activity relationships in bioactivation, different distribution processes within the body, and maybe some more yet unexplored factors.

In the present study, we aimed to elucidate the efficiency of the passage over a model for the intestinal barrier with structurally different PAs (Figure 1). To include organ-specific metabolism into this study, we analysed the metabolism rate of the intestine, liver, and lung using human microsomal fractions in order to associate possible (reactive) metabolite formation with toxic cellular effects.

## 2. Results

### 2.1. Intestinal Transfer Rate of PAs Is Structure-Dependent

The intestinal passage influences the oral bioavailability and thus the amount of a substance at the target structure. The passage of heliotrine, echimidine, senecionine, and senkirkine over the intestinal Caco-2 cell monolayer has already been published by Hessel et al. (2014) [15]. However, to get a broader overview concerning structure-dependency of the passage, we here used the additional PAs intermedine, retrorsine, lasiocarpine, and monocrotaline to complete the results with further PA-representatives from each structure type (monoester, open-chained diester and diester). As in the aforementioned study, we used the differentiated human Caco-2 cell model mimicking the small intestine, to investigate which PAs pass the intestinal barrier and to which extent.

The results of the transport assay (Figure 2) illustrated a structure-dependent passage of PAs. The monoester PAs (intermedine and heliotrine) together with echimidine showed a low recovery rate (12.3%, 32.3%, and 13.1%) in the basolateral compartment when administered apically. The open-chained diester PA lasiocarpine revealed a high passage rate comparable with the rates of the cyclic diester PAs (monocrotaline, senkirkine, senecionine, and retrorsine). The investigated PAs revealed no toxic effects in the concentration used. This was verified with cytotoxicity testing, as well as with measurements of the transepithelial electrical resistance (TEER) as indicator for monolayer integrity (Appendix A).

### 2.2. Human Intestinal Microsomes Do Not Significantly Degrade PAs

In the human intestinal tract, the expression of several xenobiotic metabolizing enzymes like CYP3A4, CYP2C9, SULT1A, and UGT1A and others were demonstrated. Caco-2 cells show less phase I enzyme activities than human intestinal tissue. Therefore, we examined the metabolism of PAs with human intestinal microsomes incubated with each PA. The remaining concentration in comparison to the initial concentration at t = 0 h was determined with LC-MS/MS.

The results in Figure 3 illustrated that the remaining PA concentrations after incubation with human intestinal microsomes compared to t = 0 h were not significantly reduced. However, some tendencies were observable, as e.g., lasiocarpine and senkirkine showed slight decreases of the concentration with considerable standard deviations.

### 2.3. Human Liver Microsomes Reduce the Concentration of PAs in a Structure-Dependent Manner

Reduction of the concentration of structurally different PAs with human liver microsomes were investigated to get more information about possible structure-activity relationships. Therefore, human liver microsomes were incubated with each PA, and the remaining PA concentrations in comparison to t = 0 h were determined with LC-MS/MS (Figure 4).

The results illustrated different reduction of the concentrations for the specific PAs. The remaining PA concentration after 4.5 h of incubation with human liver microsomes varied between 37.4% ± 10.8 for lasiocarpine and 87.3% ± 5.1 for monocrotaline. The concentration reduction seemed to be influenced by structural properties. The diester PAs echimidine, lasiocarpine, senkirkine, and senecionine demonstrated high degradation rates. Whereas the monoester PAs intermedine, and heliotrine revealed no significant reduction of the initial concentration. However, no significant reduction of the concentration was observed for the cyclic diester PAs monocrotaline and retrorsine.

### 2.4. Human Lung Microsomes Reduce the Concentration of PAs in a Structure-Dependent Manner

As secondary target organ, the lung is exposed to PAs after they passed the intestinal tract and the liver. In the lung as well as in the intestine CYP enzymes are expressed [23,24]. For this reason, reduction of the concentration of PAs with human lung microsomes was determined in order to investigate whether PAs are possibly metabolized also by extrahepatic tissues containing CYP enzymes.

The incubation of PAs with human lung microsomes revealed differences in the reduction of concentration (Figure 5). The monoester PAs intermedine and heliotrine revealed only slight reduction of the initial concentrations (89.9% ± 2.3, 71.9% ± 11.4) whereas the incubation with the diester PAs lasiocarpine, senkirkine, senecionine, monocrotaline, and echimidine demonstrated a more pronounced reduction of the PA amount in comparison to t = 0 h (55.8% ± 24.9, 56.2% ± 12.4, 70.6% ± 20.9, 75.9% ± 3.6, 67.8% ± 19.9). Even though, the reduction of senecionine is not statistically significant due to the high standard deviations.

### 2.5. Human Liver and Lung Microsomes Can Bioactivate Lasiocarpine and Monocrotaline to Induce Apoptosis in A549 Cells

Based on the result that liver and lung microsomes do efficiently metabolize lasiocarpine, we examined if both organs are able to bioactivate the PAs. Thereby we used apoptosis as marker for cellular damage in the human lung cell line A549. Lasiocarpine was chosen as one of the most hepatotoxic PA [25,26,27]. Moreover, we used monocrotaline as a well-characterized PA candidate able to induce lung damage [13,28,29]. The PAs were incubated with human liver or lung microsomes and applied to A549 cells, which were subsequently investigated regarding apoptosis.

The results illustrated that the number of cells indicating apoptotic and necrotic properties increased when incubated with liver microsomes and PAs (Figure 6A). Lasiocarpine in combination with liver microsomes induced a stronger effect than monocrotaline incubated with liver microsomes. 150 µM lasiocarpine incubated with liver microsomes induced an increase of apoptotic cells from 1.5% ± 0.8 in the control treatment to 30.3% ± 11.8. The number of apoptotic cells increased further with 250 µM lasiocarpine to 36.7% ± 11.2. Similarly, the number of necrotic cells was increased upon the incubation with 150 µM and 250 µM lasiocarpine with liver microsomes from 1.9% ± 0.9 for cells in the control treatment to 18.8% ± 11.8 and 21.7% ± 11.2, respectively. Lung microsomes together with 150 µM lasiocarpine induced an increase of apoptotic cells to 5.6% ± 3.6 and 2.3% ± 0.7 of necrotic cells. This effect was comparable to the effect of 250 µM lasiocarpine with lung microsomes (apoptotic cells: 4.2% ± 1.3 necrotic cells: 3.5% ± 1.3). The induction of Caspase 3 activity also revealed the ability of lasiocarpine to induce apoptosis when incubated with liver and lung microsomes. However, the differences in the effect strength between liver and lung microsomes are less pronounced compared to the results of the FACS measurements. The induction of Caspase 3 activity is comparable for the incubations with liver and lung microsomes. This effect can be explained with the different endpoints measured.

Lung microsomes incubated with 150 µM and 250 µM monocrotaline induced an increase of apoptotic cells in the control treatment from 1.5% ± 0.8 to 11.1% ± 6.1 and 6.1% ± 2.3 in the treated populations. The number of necrotic cells was increased similarly (3.9% ± 1.9 and 3.2% ± 3.6). The liver microsomes induced an increase of apoptotic cells (1.5% ± 0.8 to 14.9% ± 6.3) as well as an increase of necrotic cells (1.9% ± 0.9 to 6.7% ± 2.7) when incubated with 250 µM monocrotaline.

The detection of activity of Caspase 3 as indicator for apoptosis (Figure 6B) revealed a significant induction of cells incubated with lasiocarpine and liver microsomes as well as lasiocarpine and lung microsomes compared to t = 0 h. The incubation of the cells with monocrotaline and lung microsomes showed a significant increase of Caspase 3 activity in comparison to t = 0 h. However, incubation of monocrotaline and liver microsomes did not significantly induce Caspase 3 activity.

### 2.6. CYP3A4 Is the Most Important CYP Enzyme Metabolizing Lasiocarpine

Lasiocarpine and monocrotaline were incubated with human supersomes to determine which CYP enzymes are responsible for the metabolism.

The incubation of lasiocarpine (Figure 7A) and monocrotaline (Figure 7B) with the different human supersomes revealed huge differences in the metabolism rates. After 5 h incubation with CYP3A4, lasiocarpine levels were decreased up to 9%, whereas the other CYP enzymes did not significantly reduce the PA content. In contrast, none of the investigated CYP enzymes significantly decreased the levels of monocrotaline during the 5 h incubation.

## 3. Discussion

Humans and animals are exposed to PAs by contaminated food or feed. By ingestion, PAs can induce toxic effects like HSOS. Chronic exposure can result in liver haemangiosarcomas [4,9,30]. Moreover, PAs were demonstrated to induce genotoxic effects [31,32,33]. In vitro assays indicate a structure-dependency in the effect strength of PA-induced cytotoxicity and genotoxicity [22,26,27,34]. For the induction of toxic effects in a target tissue, uptake of PAs over the intestinal barrier into the body is a prerequisite. In the respective target tissue, PAs can undergo metabolism including detoxification and bioactivation reactions which are mostly mediated by CYP enzymes [4,30,35]. However, other tissues than the liver also express CYP enzymes [36]. In this study, we examined, if metabolism (including bioactivation) of eight structurally different PAs can also occur in other tissues expressing CYP enzymes possibly exposed to PAs. Moreover, the structure-dependent uptake over the intestinal epithelium was examined to complete the data.

Our study showed structure-dependent passage rates in the Caco-2 model representing the human intestinal barrier for the investigated PAs. The PAs echimidine and intermedine passed the barrier less efficiently than the other examined diester PAs (Figure 2). The differences in passage rates could indicate first structure-dependent effects explaining the observation that open-chained and cyclic diester PA induced more pronounced cytotoxic effects than monoester PAs in several studies [25,37,38]. Interestingly, the passage rates of echimidine and lasiocarpine are different even though they both belong to the open-chained diester group. This phenomenon might be explained with other structural properties than the esterification. PAs can also be grouped according their necine base. Echimidine belongs to retronecine-type PAs whereas lasiocarpine is a heliotridine-type PA. This structural feature might also influence the transport efficiency.

Thus, we conclude that one important factor in PA-mediated toxicity is the structure-dependent passage of the PAs over the intestinal epithelium since the absorption influences all following processes like the systemic transport, the body distribution, and the uptake in the liver or the metabolism.

Additionally, we showed that PAs are poorly metabolized by human intestinal microsomes (Figure 3). Investigations focussing on the metabolism of four different PA *N*-oxides (riddelliine *N*-oxide, retrorsine *N*-oxide, seneciphylline *N*-oxide, and senecionine *N*-oxide) with rat intestinal microbiota indicated that the *N*-oxides are reduced to the corresponding PA parent compounds under anaerobic conditions. Thus, the amount of ingested PA *N*-oxides might eventually add up to the ingested PAs and induce toxic effects similar to the free base PAs. However, the detection of additional metabolites revealed no detectable amount indicating that the examined microsomal preparation do not substantially contribute to the bioactivation of PAs [39]. Together with our results that the intestinal preparations induce no significant decrease of the concentration of the PAs (Figure 3), it can be assumed that no significant metabolism of PAs is mediated in the intestinal tissue.

Once having passed the intestinal epithelium, PAs are transported to the liver via the portal vein to be taken up structure-specifically as demonstrated by Enge et al. (2021). They revealed a higher uptake of diester PAs in comparison to the examined monoester PAs [16]. Moreover, other studies showed the relevance of the transporter SLC10A1 and SLC22A1 for the hepatic uptake even though it remains to be elucidated if there are also structure-specific effects. [17,18,19]. After hepatic uptake, PAs are primarily metabolized by CYP enzymes [10,12,20,21,40].

Our results demonstrated structure-dependent reduction of the concentration of PAs incubated with human liver microsomes. The highest reduction rates were observed for the diester PAs senecionine, echimidine, and lasiocarpine, whereas the investigated monoester PAs intermedine and heliotrine showed less reduction. Similar tendencies were demonstrated in other studies. Lu et al. (2020) used human hepatic endothelial cells, able to express CYP3A4, to study the effect of several structure-different PAs on the cell viability. The reported results showed that diester PAs (clivorine, retrorsine, riddelliine, senecionine, and seneciphylline) induced higher effects in reduction of cell viability than the monoester PA heliotrine and the 1,2-saturated platyphylline [41]. Moreover, Louisse et al. (2019) used HepaRG cells to demonstrate structure-specific changes in the induction of phosphorylation of the histone H2AX, as marker for genotoxicity. The diester PAs echimidine, heliosupine, intergerrimine, jacoline, and lasiocarpine exerted the highest rate of phosphorylation [38] confirming the observation of more pronounced effects by diester PAs in comparison to monoester PAs.

A closer look to the responsible enzymes for the reduction of the PA concentration upon incubation with supersomes revealed that CYP3A4 is most responsible for the metabolism of lasiocarpine (Figure 7) as already demonstrated by several studies [20,22]. However, the metabolism of monocrotaline seems to be dependent on other CYP enzymes or oxygenases (Figure 7) [10,42]. The metabolism of PAs was also shown for other species than human. Ebmeyer et al. (2019) showed the degradation of lasiocarpine by the rat orthologous enzymes Cyp3a1 and Cyp3a2. They demonstrated an efficient degradation upon incubation with supersomes containing these enzymes [20]. Kolrep et al. (2018) investigated differences in the degradation of PAs upon incubation with S9 fractions from different animals. This study showed that different species show different susceptibility to metabolize PAs [43]. However, this study covers only one part of the toxicokinetic of PAs. The uptake and transport must also be considered when defining PA-induced toxicity in animals.

Even if the liver is known as the main xenobiotic-metabolizing organ, CYP enzymes are also expressed in other organs like the lung or the intestinal tract [44]. In the lung, for example, the CYP enzymes 1A1, 1A2, 2A6, 2B6, 2C8, 2C18, 3A4, 3A5 and others were found to be expressed [23]. The intestinal tract was reported to express CYP1A1, 1B1, 2C9, 2C19, 2D6, 2E1, 2J2, 2S1, 3A4, and 3A5 [23]. However, our results indicated that human intestinal microsomes did not substantially reduce the PA concentrations (Figure 3). Moreover, the same experiment with human lung microsomes illustrated that only the concentrations of lasiocarpine, senkirkine, and echimidine were reduced (Figure 5). These findings suggest that most of the metabolism of the investigated PAs seem to be located in the liver. However, the concentration of monocrotaline incubated with liver microsomes was slightly but not significantly decreased (Figure 4).

Nevertheless, monocrotaline is used frequently as a substance to induce pulmonary arterial hypertension in animals [4,45,46]. The detection of pyrrole-protein adducts after administration of different PAs, among them monocrotaline, to rats showed that monocrotaline induced the highest lung/liver pyrrole-protein adduct ratio. This phenomenon shows that in the case of monocrotaline more metabolites were transported to the lung than it is the case for other PAs [47]. This observation shows that monocrotaline might affect the lung more than the liver. Nevertheless, there are studies demonstrating that PA-induced pneumotoxicity in rats depend on prior bioactivation of the PAs in the liver [12,48]. The fact that pneumotoxicity was demonstrated in several studies even though the metabolites were reported to be unstable [49] was explained with the stabilisation of the metabolites with red blood cells [48,50]. Conclusively, there are several hints that PAs, and especially monocrotaline, are metabolized in the liver and the metabolites are transported to the lung.

Nevertheless, our experiments also demonstrated that the metabolism of monocrotaline by human liver and lung microsomes induced apoptosis in human lung A549 cells (Figure 6) even though the assays revealed differences in the effect strength. These differences can be explained with the different endpoints that are detected. Caspase 3 is responsible for the cleavage of key proteins initiating controlled cell death [51]. Therefore, the detection of an increased Caspase 3 activity specifically detects the induction of apoptosis whereas other factors like the part of viable cells are not detected, whereas in the FACS measurements, the entire cell population is labelled and sorted into viable, necrotic, and apoptotic cells. Moreover, the assays might have differences in the sensitivity of the detected effects. These reasons might explain the differences in the results and effect strengths. Noteworthy, the assays both indicate the ability of lasiocarpine and monocrotaline to induce apoptotic cell death when incubated with liver or lung microsomes. This result seems to be contradictory to the observation, that PA metabolites are transported from the liver to the lung. Nevertheless, it remains to be elucidated if the effects can appear together and add up to the observed pneumotoxic effects or if another, yet unexplored factor, also affects the PA-mediated pneumotoxicity.

Taken together, the passage of PAs from the intestinal tract into the systemic circulation system showed significant differences for the investigated PAs (monoester: intermedine and heliotrine, open-chained diester: echimidine and lasiocarpine, cyclic diester: monocrotaline, senkirkine, senecionine, and retrorsine) (Figure 2). Moreover, the bioactivation rate of PAs with human liver microsomes was also highly structure-dependent (Figure 3). The combination of this information together with already published in vitro data concerning the structure-dependent cytotoxicity and uptake could further improve the knowledge of molecular processes leading to observed effects of PAs in vivo [16,22,25,52]. In fact, the results observed in the present study can help to explain effects seen in in vivo studies. He et al. (2021) explored the influence of hepatic and intestinal CYP enzymes towards the bioactivation of retrorsine in mice. Thereby, the suppression of intestinal CYP activities resulted in an unchanged PA metabolism indicating that the hepatic CYP enzymes play the main role in PA bioactivation. Interestingly, pyrrole-protein adducts were identified in the intestine of the mice, indicating a transport of PA metabolites from the liver back to the intestine [53]. Another study with mice detected the transport of PA metabolites to the lung exerting toxic effects there [12]. In combination with our results, these observations strongly indicate that most of the PAs are bioactivated in the liver and then transported to other organs like the lung.

Moreover, the demonstrated structure-dependencies are in line with several other studies [25,38,41,54] and can be the basis for an adapted risk assessment taking these effects into account. Therefore, the results give valuable information about human toxicokinetics indicating strong structure-dependencies. Since the toxicokinetic parameters finally influence the toxicity, our results give hints regarding the effect strength as well. Moreover, these data can be used as basis for new approach methodologies (NAM), for example for the application in physiologically based kinetic (PBK) modelling.

## 4. Materials and Methods

### 4.1. Chemicals

PAs (purity > 95 %) were purchased from PhytoLab (Phytolab GmbH & Co. KG, Verstenbergsgreuth, Germany). PAs were dissolved in 50% (*v*/*v*) water/acetonitrile (ACN) in stock solutions of 5 mM.

Dulbecco´s Modified Eagle Medium (DMEM) and fetal bovine serum (FBS) were purchased from Pan-Biotech (Pan-Biotech, Aidenbach, Germany). Penicillin and streptomycin were purchased from Capricorn (Capricorn Scientific, Ebsdorfergrund, Germany). If not stated otherwise, all other chemicals were purchased from Sigma Aldrich (Sigma Aldrich, Taufkirchen, Germany).

7-AAD was purchased from Invitrogen AG (Carlsbad, CA, USA). Annexin-V-FITC was purchased from Enzo Life Sciences GmbH (Lörrach, Germany).

All assays were performed with representatives of monoester (intermedine and heliotrine), open-chained diester (echimidine and lasiocarpine), and cyclic diester (monocrotaline, senkirkine, senecionine, and retrorsine) PAs.

### 4.2. Cell Culture

#### 4.2.1. Caco-2 Cells

Caco-2 cells were purchased from the European Collection of Cell Culture (Salisbury, United Kingdom). Caco-2 cells are commonly used as a model for the small intestine due to differentiation into a monolayer exhibiting structural and biochemical characteristics similar to human enterocytes [55,56]. The cells were maintained at 37 °C, with 5% CO_2_, in humified atmosphere in DMEM supplemented with 10% (*v*/*v*) FBS, 100 U/mL penicillin, and 100 μg/mL streptomycin. The cells were seeded in different plate formats depending on the assays performed. Transport experiments were conducted as described by Hessel et al. (2014) [15]. Briefly, 60,000 cells were seeded in Transwell inserts with 0.4 µm pore size polycarbonate membranes (1.12 cm^2^ growth area, Corning B.V. Life Sciences, Amsterdam, The Netherlands) and were allowed to differentiate for 21 days before PA treatment. During this time, the cell culture medium was changed every day. TEER measurements (EVOM—volt ohmmeter with chopstick electrode; World Precision Instruments, Sarasota, FL, USA) were performed before and after each experiment to verify that the cell monolayer remained intact. The Caco-2 monolayers were verified to have TEER values higher than 600 Ω × cm^2^.

For the transport experiment, 0.25 µM of the respective PA was added on the apical side and cell culture medium samples were taken after 0, 8, 24, and 48 h from the respective compartment during the incubation for subsequent PA quantification via liquid chromatography-mass spectrometry (LC-MS/MS).

Cytotoxicity was detected with the WST-1 assay (Roche Diagnostics GmbH, Rotkreuz, Switzerland). 5000 cells per well were seeded in 96-well plates and allowed to differentiate for 21 days. Next, they were iterated with 200 µL of 0.1 or 1 µM of each PA and incubated for 24 h. Afterwards, 20 µL of WST-1 solution was added and cells were incubated at 37 °C for another 20 min. The absorption was detected with a TecanM200Pro spectrometer (Tecan Group Ltd., Männedorf, Switzerland) at λ = 450 nm.

#### 4.2.2. A549 Cells

The alveolar basal epithelial adenocarcinoma cell line A549 was used as model for human lung endothelial cells. The cells were cultivated in DMEM supplemented with 10% (*v*/*v*) FBS, 100 U/mL penicillin, and 100 μg/mL streptomycin. 50,000 cells/well were seeded in 12-well plates for the detection of plasma membrane asymmetry with flow cytometry and allowed to grow for 24 h before incubation. For the determination of the Caspase 3 activity, 10,000 cells/well were seeded in 96-well plates and allowed to grow for 24 h.

#### 4.2.3. Incubation of PAs with Microsomes and Recombinant Human Microsomes

Lung and intestinal microsomes were purchased from SEKISUI XenoTech, LLC (Kansas City, KS, USA). Liver microsomes were purchased from Corning (Corning Inc., New York City, NY, USA). Recombinant human microsomes (supersomes) were purchased from Becton Dickinson (Heidelberg, Germany). The supersomes allow the identification of the role of specific CYP enzymes (CYP1A1, CYP1A2, CYP2A6, CYP3A4, CYP3A5) in the metabolism of the investigated PAs.

PA stock solutions were diluted with DMEM to a concentration of 12.5 µM to reach a final concentration of 10 µM when incubated with microsomes. All microsomes were diluted to a concentration of 2 mg/mL as described by Ebmeyer et al. (2019) [20]. PAs were incubated with 20% of the specific microsomes at 37 °C for 0, 1, 2 or 4.5 h. At each time point, 50 µL of each sample was transferred to 76% methanol in DMEM to stop the enzymatic reaction and precipitate proteins. This solution was centrifuged at 4 °C for 10 min and 12,000× *g*. The supernatant was diluted and prepared for the LC-MS/MS measurements.

#### 4.2.4. LC-MS/MS

The determination of PA concentrations after incubation with microsomes or supersomes was performed by LC-MS/MS as described earlier by Kaltner et al. (2019) and Enge et al. (2021) [16,57]. Briefly, a 50 × 2.1 mm Kinetex 2.6 μm Core-Shell EVO C18 100 Å column (Phenomenex, Aschaffenburg, Germany) protected by a SecurityGuard ULTRA EVO C18 2.1 mm guard column (Phenomenex, Aschaffenburg, Germany) was used for chromatographic separation on a Shimadzu Prominence HPLC device (LC-20AB, SIL-20AC HT, CTO-20AC, CBM-20A, Shimadzu, Duisburg, Germany). Column oven temperature was maintained at 30 °C, the flow rate was consistently hold at 0.4 mL/min and the injection volume was 10 µL. The HPLC system was coupled to an API4000 triple quadrupole MS (Sciex, Darmstadt, Germany) which was operated in positive electro spray ionisation (ESI) mode with the following parameters: ionisation voltage: 2500 V; nebuliser gas: 50 psi; heating gas: 50 psi; curtain gas: 30 psi; temperature: 600 °C; collision gas: level 7. The selected PA analytes were determined in multiple reaction monitoring (MRM) mode and quantified by external calibration standards ranging from 10 to 125 nmol/mL in DMEM. The PA content was always normalized to the content at t = 0 h.

### 4.3. Detection of Apoptosis

#### 4.3.1. Plasma Membrane Asymmetry Detection with Flow Cytometry

The detection of an unusual distribution of phosphatidylserine (PS) in the outer plasma membrane as indicator of early stages of apoptosis was performed using flow cytometry. PS was stained specifically by Annexin V. The discrimination between early-stage apoptotic cells and dead/necrotic cells was verified with the staining of nucleic acids by 7-amino-actinomycin D (7-AAD). This dye enters into cells only via damaged plasma membranes. Conclusively, the discrimination between early apoptotic, dead/necrotic, and viable cells is possible. Cells (50,000 per well) were seeded in 12-well plates in a final DMEM volume of 1 mL. Cells were treated with PAs and human liver or lung microsomes. After incubation at 37 °C for 4 h, the microsomal incubation solution was removed, and cells were washed with DMEM. An incubation for further 20 h followed before the change of DMEM and addition of FasL ligand (Anti-human CD95, FAS 18, ImmunoTools GmbH, Friesoythe, Germany) to prevent an unspecific activation of the Fas receptor. After a second incubation step for 24 h, cells were washed with 500 µL PBS and trypsinized with 350 µL trypsin/EDTA. Upon addition of 500 µL DMEM, cells were centrifuged for 3 min at 500× *g* and the supernatant was discarded. The cell pellet was washed with 500 µL PBS. After an additional centrifugation with the same parameters, the supernatant was removed before the cells were washed with 500 μL Annexin-V-buffer (10 mM 2-[4-(2-hydroxyethyl)piperazin-1-yl]ethanesulfonic acid (HEPES), 140 mM NaCl, 5 mM CaCl_2_ × 2 H_2_O, pH 7.4). Staining was performed by the addition of 50 µL staining solution (87.5% (*v*/*v*) Annexin-V-buffer, 10% (*v*/*v*) 7-AAD, and 2.5% (*v*/*v*) Annexin-V-FITC) on ice for 20 min in the dark. Next, fluorescent signals were analyzed with the flow cytometer BD Accuri C6 (BD Biosciences, Erembodegem, Belgium). Thereby, 20,000 events were detected in each measurement at a flow rate of 66 µL/min.

#### 4.3.2. Caspase 3 Activity

A specific fluorogenic substrate (Ac-DEVD-AFC, TIB MOLBIOL GmbH, Berlin, Germany) was used to determine Caspase 3 activity as indicator of apoptosis. The cleavage of this substrate was measured fluorometrically. For this purpose, 10,000 A549 cells per well were incubated with the PAs in 96-well plates for 4 h at 37 °C. Subsequently, DMEM was removed, and the cells were washed twice with DMEM. Afterwards, the cells were incubated for additional 20 h. After a total of 24 h incubation, cells were lysed at room temperature (RT) for 20 min with 50 μL lysis buffer (50 mM HEPES, 2% (*v*/*v*) Triton-X-100). This was followed by a 20 min incubation at RT while shaking. Next, 5 μL 1 M dithiothreitol (DTT) and 0.5 μL of the fluorogenic substrate were added to 94.5 μL reaction buffer (50 mM HEPES, 5% (*v*/*v*) glycerin, 0.1% (*w*/*v*) 3-[(3-cholamidopropyl)dimethylammonio]-2-hydroxy-1-propanesulfonate (CHAPS), 5 mM 2,2′,2″,2‴-(ethane-1,2-diyldinitrilo)tetraacetic acid (EDTA)) per well. Afterwards, the fluorescence intensity was recorded with a TecanM200Pro spectrometer (Tecan Group Ltd., Männedorf, Switzerland) at λ_ex_ = 380 nm and λ_em_ = 500 nm. Values were compared to the respective control incubations with medium and liver or lung microsomes but without PAs.

#### 4.3.3. Statistics

Statistical analysis was performed with SigmaPlot 14.0 (Systat Software, Erkrath, Germany). All assays were performed in three individual experiments. The generated data were analyzed regarding statistical differences in comparison to the negative control or t = 0 h with One Way analysis of variance (ANOVA) followed by Dunnett’s test. The significance levels were set as * *p*< 0.05, ** *p*< 0.01, *** *p*< 0.001.

## Figures and Tables

**Figure 1 ijms-23-09214-f001:**
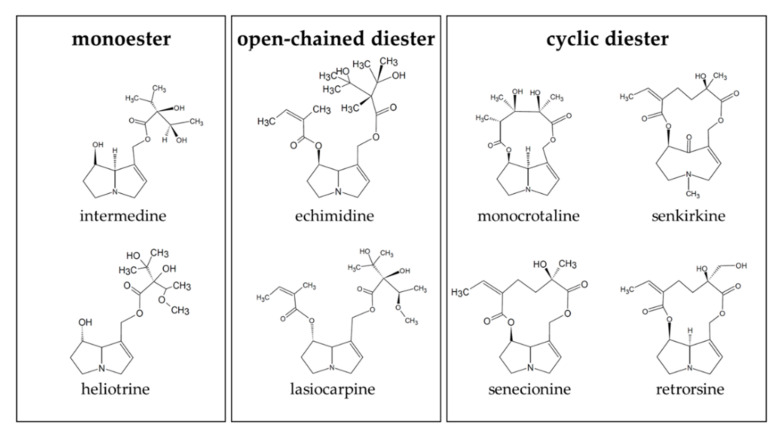
Structures of the PAs used in this study. The PAs were chosen to represent the different structures: monoester (intermedine and heliotrine), open-chained diester (echimidine and lasiocarpine) and cyclic diester (monocrotaline, senkirkine, senecionine, and retrorsine).

**Figure 2 ijms-23-09214-f002:**
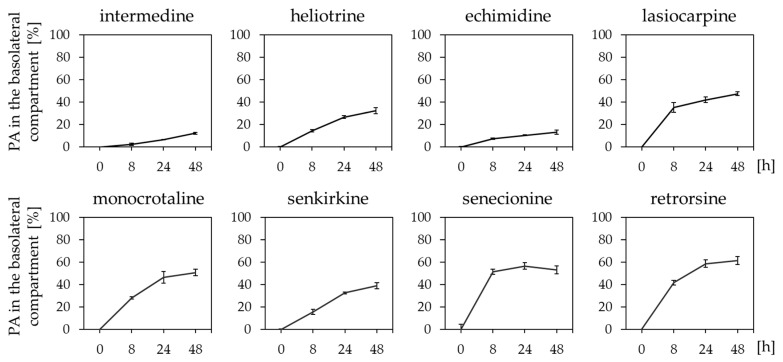
Transfer rates of structurally different PAs over the differentiated Caco-2 cell monolayer. Caco-2 cells were seeded and differentiated in Transwell inserts and incubated with 0.25 µM of the specific PAs from the apical compartment of the Transwell chamber. At the indicated time points, the basolateral PA amounts were determined with LC-MS/MS. Mean values ± SD of the PA concentrations in the basolateral compartment in % in comparison to the applied PA concentration are shown from three individual experiments.

**Figure 3 ijms-23-09214-f003:**
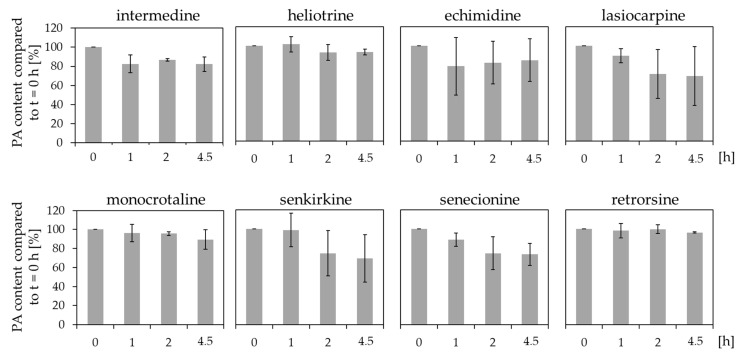
Reduction of the concentration of PAs after incubation with human intestinal microsomes. 10 µM PAs were incubated with microsomes at 37 °C for up to 4.5 h. At the indicated time points, the remaining PA concentration in comparison to t = 0 h was determined. For each incubation three individual experiments were performed. Mean values ± SD are shown. Statistical differences to t = 0 h were determined with One-Way ANOVA followed by Dunnett’s post hoc test.

**Figure 4 ijms-23-09214-f004:**
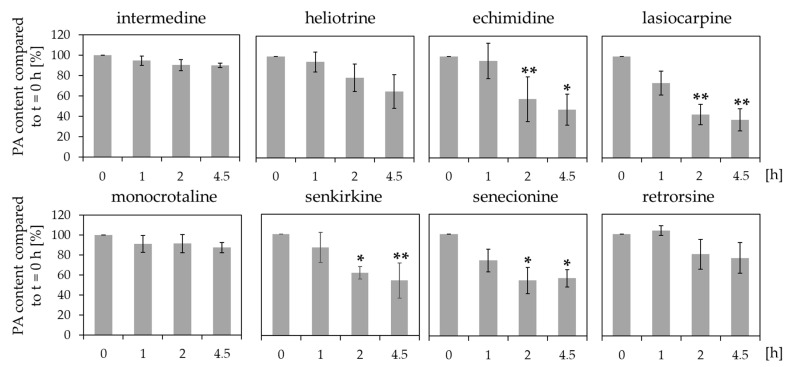
Reduction of PA concentrations after incubation with human liver microsomes. 10 µM PAs were incubated with liver microsomes at 37 °C for up to 4.5 h. At the indicated time points, the remaining PA concentration in comparison to t = 0 h was determined. For each incubation three independent experiments were performed. Mean values ± SD are shown. Statistical significance was determined with One-Way ANOVA followed by Dunnett’s post hoc test and is indicated as * *p* < 0.05, ** *p* < 0.01.

**Figure 5 ijms-23-09214-f005:**
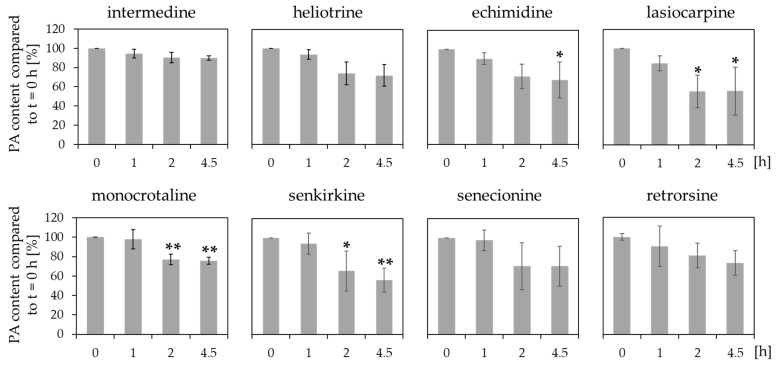
Reduction of concentration of PAs after incubation with human lung microsomes. 10 µM PAs were incubated with human lung microsomes at 37 °C for up to 4.5 h. At the indicated time points, the remaining PA concentration compared to t = 0 h was determined. For each incubation three independent experiments were performed. Mean values ± SD are shown. Statistical significance was determined with One-Way ANOVA followed by Dunnett’s post hoc test and is indicated as * *p* < 0.05, ** *p* < 0.01.

**Figure 6 ijms-23-09214-f006:**
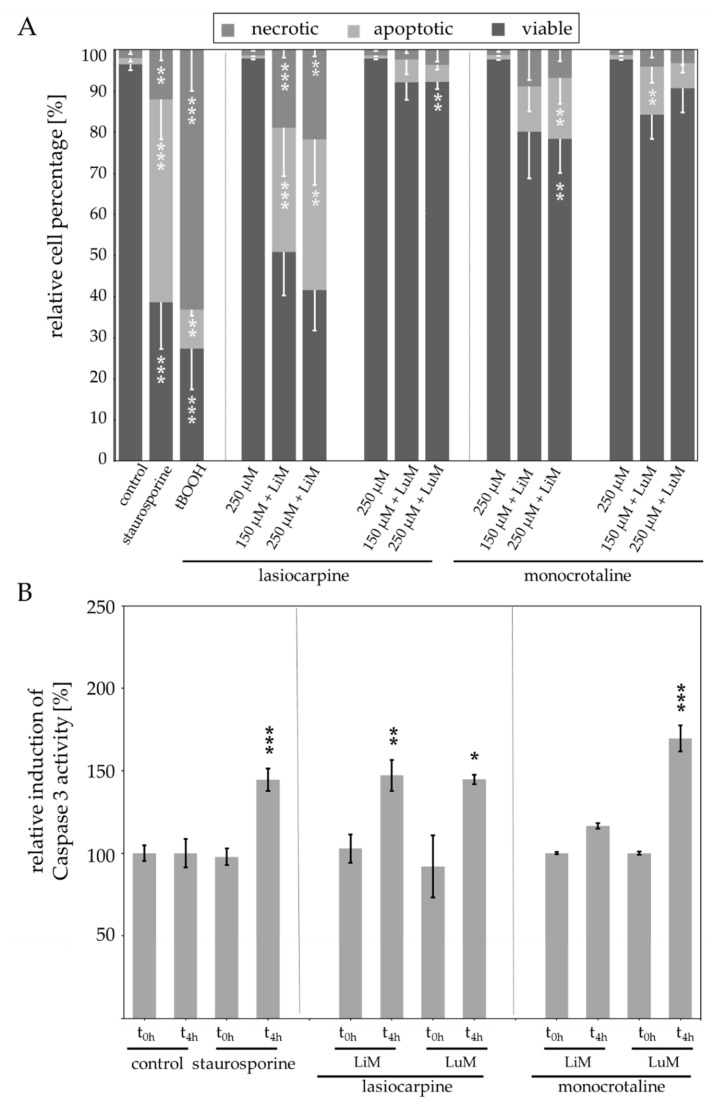
Induction of apoptosis in A549 cells upon incubation with bioactivated PAs: (**A**) Induction of apoptosis in A549 cells after incubation with 150 µM or 250 µM monocrotaline or lasiocarpine for 4 h + 20 h. As external metabolism system human liver (LiM) or lung (LuM) microsomes were used. Cells were incubated with microsomes (diluted to a final concentration of 2 mg/mL) and PAs for 4 h before washing and incubating for additional 20 h for a total time of 24 h. Induction of apoptosis was measured by flow cytometry by detection of the fluorescence intensities of Annexin-V-FITC and 7-aminoactinomycin (7-AAD). The cells were incubated with medium as negative control and staurosporine as positive control inducing apoptosis or tBOOH as another positive control inducing necrosis. (**B**) Alteration of Caspase 3 activity in A549 cells upon incubation with 250 µM lasiocarpine or monocrotaline activated priorly with human liver or lung microsomes for 24 h. Cells were incubated with PAs and microsomes for 4 h before washing and further incubation for 20 h. Caspase 3 activity was measured by adding the substrate Ac-DEVD-AFC after lysis of the cells and photometric detection of emission at different time points. For each incubation, three independent experiments were performed. Mean values ± SD are shown. Statistical significance in comparison to the medium control with the respective microsomes but without PAs was determined with One-Way ANOVA followed by Dunnett´s post hoc test and is indicated as * *p* < 0.05, ** *p* < 0.01, *** *p* < 0.001.

**Figure 7 ijms-23-09214-f007:**
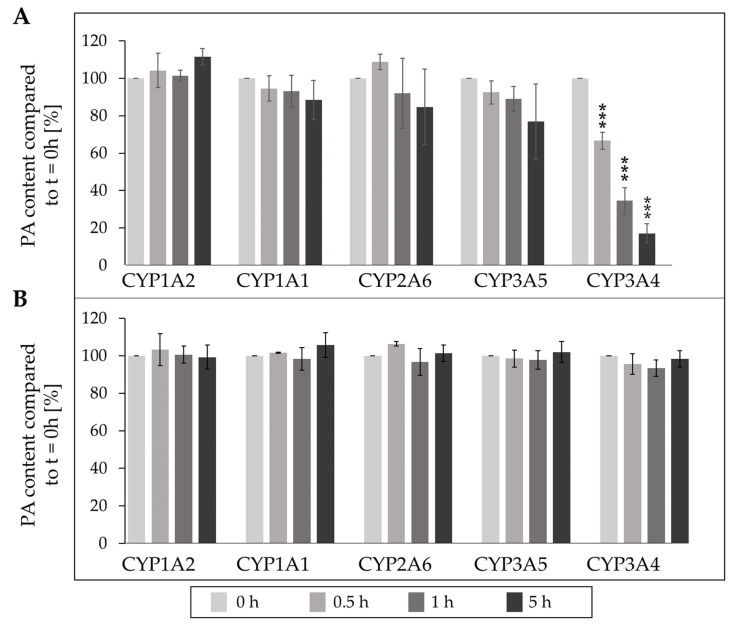
Reduction of concentration of lasiocarpine (**A**) and monocrotaline (**B**) upon incubation with human supersomes. 10 µM of respective PA were incubated at 37 °C with the CYP supersomes for up to 5 h. At the indicated time points, samples were taken, and the enzymatic reaction was stopped with methanol. The remaining PA concentration was then determined with LC-MS/MS. Three independent experiments were performed for each incubation. Mean values ± SD are shown. Statistical significance was determined with One-Way ANOVA followed by Dunnett´s post hoc test and is indicated as *** *p* < 0.001.

## Data Availability

The data presented in this study are available on request from the corresponding author.

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
