# Peer review of "Structure-Dependent Toxicokinetics of Selected Pyrrolizidine Alkaloids In Vitro"

_ijms, 2022, doi:10.3390/ijms23169214_

Round 1
Reviewer 1 Report
In this manuscript, the authors investigated the transport and metabolism rates of structurally different pyrrolizidine alkaloids (PAs) using a model of the intestinal barrier and human microsomes prepared from the intestine, liver, and lung. Subsequently, cytotoxic effects of two PAs, lasiocarpine and monocrotaline, in the presence or absence of human liver or lung microsomes on A549 cells were investigated. Although studies on toxicokinetics of PAs may be important in toxicology, this manuscript seems to be insufficient for publication in this journal as pointed out below.
Major comments
1. In Figure 2, lasiocarpine passed the caco-2 cell monolayer much more efficiently than echimidine, even though they belong to the same group (open-chained diester). However, there is no discussion about this point. Because I think readers will be curious about the point, the authors should discuss the difference in the transport rates between them. In addition, are there differences in toxicity in vivo between them? If so, can it be explained by the difference in the transport rates?
2. No discussion is made about the lack of parallel results in Figures 6A and 6B. First, lasiocarpine strongly induced apoptosis of A549 cells in the presence of human liver microsomes (LiM) but not lung microsomes (LuM) in Figure 6A, whereas no difference in caspase 3 activity was observed between the two groups in Figure 6B. Second, monocrotaline (250 μM) significantly induced apoptosis in the presence of LiM but not LuM in Figure 6A, whereas caspase 3 activity was markedly increased by monocrotaline in the presence of LuM but not LiM in Figure 6B. Third, lasiocarpine (250 μM) in the presence of LiM induced apoptosis more strongly than monocrotaline (250 μM) in the presence of LuM in Figure 6A, whereas the caspase 3 activity of the cells incubated with monocrotaline in the presence of LuM was higher than that with lasiocarpine in the presence of LiM in Figure 6B. I think readers may not be convinced of such non-parallel results.
3. In Figure 6, the effect of microsomes alone (without PAs) on apoptosis has not been shown. Did the authors confirm it in the first place? If so, the results should be described. Additionally, the final concentrations of microsomes (e.g., mg protein/mL) in the medium should be written in the figure legend and the corresponding ‘Materials and Methods’ sections.
4. In Figure 7, there is no explanation why these five P450 enzymes were selected. Additionally, it is desirable to show that these supersomes are working fine with positive control substrates (e.g., phenacetin for CYP1A2).
Minor comments
1. I cannot understand the meaning of [%] PA on the vertical axis label in Figure 2. Please explain it in the figure legend. In addition, SDs (error bars) are not shown in Figure 2, so please add them.
2. In Figure 5 (senecionine and retrorsine), ‘0 h’ columns are not set to 100%, so please correct.
3. The resolution of Figure 6 is poor, so please improve it. Additionally, there was no description about staurosporine and tBOOH. The authors need to explain what these are and the results of treatment with them. Further, how much does staurosporine increase caspase 3 activity? Please show if the data exists.
4. Page 8, line 223, “lasiocarpine levels were decreased up to 91 %” should be corrected to “lasiocarpine levels were decreased up to 9 %”.
5. Page 11, line 384, the sentence ends halfway. Please correct.
6. Page 12, line 392, although it is written as “PA stock solutions were diluted with DMEM to a final concentration of 12.5 µM”, isn't it a mistake of 10 µM?
7. Page 12, line 425-427, it's just a line of words, not a sentence, so please correct.
8. There are many mistakes such as typographical errors and grammatical errors throughout the manuscript, so the authors should proofread the sentences again.
Reviewer 2 Report
July 14, 2022
Editors “International Journal of Molecular Sciences”
Dear Editors,
Thank you for the opportunity to review your manuscript entitled “Structure-dependent toxicokinetic of pyrrolizidine alkaloids”. This is a well written paper that with minimal modification is deserving of publication. I do have a couple suggestions that might make it a better work. Hopefully the authors will use these suggestions as constructive suggestions to improve and clarify this excellent work.
Specific Suggestions
Title- “Toxicokinetic” is probably not accurate as more than one process is present. The pleural “toxicokinetics” might be better as they studied the movement and metabolism of these toxicants.
“Pyrrolizidine alkaloids” suggests all were included when only a selected few were studied.
All these were “in vitro” analyses were done using human cells, microsomes, supersomes etc. Nearly all other work has been done using animals or animal products. The title might be improved by reflecting this.
Abstract- The purpose or objectives of these studies might be directly stated. The last sentence is a pretty good summary, but it might be more direct to state the objective for these in vitro models.
The conclusion could use some focus. A summary of absorption, metabolism and subsequent cytotoxicity with comparison with animal and other cytologic and toxicologic studies might contribute. How do these correlate with reported dehydro-pyrrolizidine alkaloid toxicity?
There is no mention of the CYP3A4 work.
What is the picture for human toxicokinetics, toxicity and possibly carcinogenicity?
Introduction- This section is nicely done and well referenced.
Results- Figure 2 may be incomplete. It suggests the assays were done in triplicate and the data is presented as means ± standard deviations. However, line graphs are presented with no error bars. The y axis is poorly labeled and the tick marks do not correlate with the times.
Apically might be confusing as it generally refers to forming an apex or at the tip.
TEER might need some explanation.
tOh might also be confusing.
Figure 3 legend includes the set description of the statistical analysis, but as there was no differences between times nothing is indicated on the graphs. Perhaps a statement that there were no significant differences would prevent readers from searching for those results.
L119 dependnet is misspelled.
Figures and their legends are supposed to be able to understandable. Some explanation of the control, staurosporine and tBOOH may be indicated. Determination of necrotic, apoptotic and viable cells is poorly linked to the legend description of the flow cytometry.
Purified dehydro-pyrrolizidine alkaloids (DHPAs) were compared by absorption, human intestinal microsome metabolism, human liver microsome metabolism, and human lung microsome metabolism. This is wonderful work and a direct comparison of alkaloids’ responses in these different assays with the reported toxicities in reported animal models would be very informative. It may also be useful in identifying where differences in human toxicokinetics might be further investigated. A numerical table might be illustrative.
Discussion- L229 This needs a little better explanation. Chronic riddelliine intoxicatioin causes liver hemagiosarcomas and other vascular lesions in rats and mice, but that is not the only DHPA-induced neoplasm, especially at different doses and durations.
L248 There is some disconnect here as N-oxide metabolism is not compared in figure 3 in your intestinal absorption model. Intestinal reduction of these N-oxides clinically and experimentally seems to suggest that with oral exposure have toxicity similar to the free base DHPAs. In any case this discussion probably should be referenced.
L281 Human CYP3A4 DHPA metabolism might be briefly compared with its contribution to metabolism in other species.
L301 then should probably be than
